# Low Cholesterol Levels in Younger Heart Failure Patients May Predict Unfavorable Outcomes

**DOI:** 10.3390/medicina59071314

**Published:** 2023-07-16

**Authors:** Lior Charach, Itamar Grosskopf, Leonid Galin, Irit Guterman, Eli Karniel, Gideon Charach

**Affiliations:** 1Department of Internal Medicine B, Meir Medical Center, Kfar Saba 4428164, Israel; 2Sackler Faculty of Medicine, Tel Aviv University, Tel Aviv 6997801, Israel

**Keywords:** cholesterol, heart failure, low density lipoprotein, statins

## Abstract

*Background and Objectives:* Hypercholesterolemia is a main risk-factor leading to ischemic heart disease (IHD). However, among patients with heart failure, the use of lipid lowering drugs in the presence of low cholesterol might be dangerous. This 18-year longitudinal study of patients ≤51 years old investigated the relationship between baseline total cholesterol, low-density lipoprotein cholesterol (LDL-c) and triglyceride levels, and survival among patients with severe HF. *Materials and Methods:* The average NYHA score of 82 patients ≤51 years old with heart failure was 2.61. They were followed for a mean of 11.3 years (15 months–20 years). Total mortality was 22%. Patients were divided into three groups. Group 1 had plasma LDL-c levels ≤ 80 mg/dl, Group 2, 80–115 mg/dl and Group 3 > 115 mg/dl. *Results:* Patients with the highest baseline total cholesterol, triglyceride and LDL-c levels > 115 mg/dl had a better survival rate (83%) compared to those with LDL-c < 80 mg/dl (50% survival, *p* = 0.043). The association between higher LDL-c levels and lower mortality was most noticeable among patients with heart failure. *Conclusion:* Longitudinal follow-up found that low LDL-c levels may indicate poorer prognosis among patient with heart failure who are ≤51 years old, similar to elderly heart failure patients. Cholesterol lowering drugs in younger patients with heart failure may increase mortality.

## 1. Introduction

It is well-established that lower serum cholesterol level is associated with better outcome among patients with coronary artery disease (CAD) and is a predictor of poorer outcome among patients who do not receive statin treatment for CAD [1,2,3,4,5,6]. Lipid levels, total cholesterol and, especially, low-density lipoprotein cholesterol (LDL-c) are major risk-factors for CAD and stroke, and are associated with increased mortality [1,2]. Primary and secondary prevention of cardiovascular events is based on treatment with statins [3,4,5]. However, studies describing decreasing LDL-c in patients with heart failure (HF) are contradictory. The consensus is that increased mortality is associated with high LDL-c levels [1,2,6]. However, to decrease cholesterol levels, some studies [7,8,9] recommended prescribing statins for patients with HF. Statins improve endothelial dysfunction and decrease C-reactive protein (CRP), which reflects inflammatory processes, oxidative stress and oxidized LDL levels in atheroma, all of which are involved in HF and complicate its pathogenesis [8,9,10]. Despite this, other studies have reported paradoxical results of decreased total and LDL-c (especially through statin treatment), indicating that they might be harmful for patients with HF [1,5,9,10].

Ubiquinone (Q10) is an essential product of the cardiac mitochondrial respiratory chain. It is involved in producing ATP by allowing the anti-oxidant effects and is lower among individuals with congestive HF [8,9,10,11,12,13,14]. Statins have been reported to decrease Q10 concentrations, which is important in cardiac myofibril-mitochondrial respiratory reactions. It also increases ATP production and potentiates anti-oxidant effects, which are diminished in patients with HF [11,15,16]. Thus, statin treatment could potentially be hazardous. Furthermore, total cholesterol, LDL-c and triglyceride-rich particles can neutralize microbial endotoxins that induce inflammation and cytokine and interleukin secretion [10,11]. Low cholesterol concentrations increase the predisposition to infections, which occur frequently among HF patients [6,7,8]. On the other hand, other studies [5,6,7,8,9,11,12,13,14,15,17,18,19,20,21,22] and case reports found an improved prognosis among patients with HF who were treated with statins. The CORONA study [2,17] did not find that rosuvastatin had a positive effect on cardiovascular mortality from various causes, myocardial infarction and nonfatal stroke in elderly HF patients with reduced ejection fraction. On autopsy, CAD was found in 33% of patients who died from pump failure. Mortality from HF was not improved by rosuvastatin.

Compared to younger individuals, people aged 65 and older are much more likely to have a myocardial infarction or stroke, develop heart disease or especially HF, which is mostly a complication of CAD and myocardial infarctions; most often termed cardiac syndrome [5,6,7,8,9,11,12,13,14,15,16,17,18,19,20,21,22].

This 18-year, historical cohort study investigated baseline low LDL-c concentrations as an unfavorable prognostic factor among a cohort of patients who were at the relatively young ages of <51 years old, who had moderate-to-severe HF, undergoing follow-up in a HF unit. 

## 2. Materials and Methods

Consecutive patients <51 years-old who were patients in the outpatient HF clinic at the Sourasky Medical Center in Tel Aviv, were recruited for this study. Baseline blood tests from 1/1998 to 7/2001 for hemoglobin levels, lipid profiles and kidney function were obtained. Exclusion criteria were metastatic cancer, severe cerebrovascular disease, end stage renal failure acute inflammatory disease and advanced dementia.

Patients were diagnosed with systolic HF based on echocardiography or Tc99 scan ventriculography results of <40% left ventricular ejection fraction (LVEF). A cardiologist examined all patients at their initial visit. Information on medications, medical history, weight, pulse, blood pressure, echocardiogram, ventricular Tc99 scan and New York Heart Association (NYHA) class was obtained. They were followed by a cardiologist from the HF unit followed the patients at least every 3 months or more often, if required. The study endpoint was all-cause mortality. 

### 2.1. Ethics Committee Approval

The Ethics Committee of Tel Aviv Medical Center (0554-17-TLV) approved the study protocol. It was registered at Clinical Trials.gov (NCT01601444). All participants provided Written informed consent was obtained from all participants before data collection.

### 2.2. Data Analysis

Nominal variables are presented as numbers and percentages. Continuous parameters are described with means and standard deviations. Chi Square or Fisher’s exact test was used to evaluate qualitative data to determine differences among the three study groups. Student’s *t*-test was used for two group comparisons. The three groups of LDL levels were compared using one-way ANOVA. A Shapiro–Wilk test was used to check for normal distribution among the continuous parameters. Mann–Whitney or Kruskal–Wallis tests were used when data were not normally distributed. Post hoc with Bonferroni Correction was used to determine differences between the 3 LDL groups. Kaplan–Meier survival analysis was used to evaluate differences in survival between the three groups of LDL cholesterol and two groups of triglycerides. Cox regression analysis was used to find parameters that independently explained differences between the dependent variables. *p*-values < 0.05 indicated statistical significance. SPSS-28 was used to analyze the data (IBM Corp., Armonk, NY, USA).

## 3. Results

From January 1998 to July 2001, 97 consecutive patients in the HF unit, aged 51 years or younger, met the inclusion and exclusion criteria and were enrolled in the study. Subsequently, 13 were excluded due to noncompliance or missing data. The final study cohort included 84 patients. A Median follow up was 11.3 years. HF duration ranged from 19 months to 18 years. Mean age was 48.3 ± 3 years (range 40–51), 65% were males and 74% had ischemic cardiomyopathy. Among the remaining 26% of patients, 9% had aortic stenosis of atherosclerotic etiology, 6% had mitral valve regurgitation related to left ventricular dilatation (ischemic cardiomyopathy) and 3% had tricuspid valve insufficiency as a result of left HF—all of which developed due to ischemic heart disease. All patients had HF with reduced ejection fraction. The mean NYHA class was 2.61 and the mean LVEF was 36.5%. Mean weight was 84 kg. The mean number of visits to the HF unit during follow-up was 24.3 (range 15–35). Type 2 diabetes mellitus was prevalent in 30%, 46 (54.8%) had hypertension (HTN) and 32 (37.6%) smoked. Women had a lower ejection fraction compared to men (33.5% vs. 36.4%, respectively; *p* = 0.027).

To determine the LDL value that would predict mortality, the study patients were divided into three groups based on the level of LDL-c level: ≤80 mg/dl for Group 1 (*n* = 15), 80–115 for Group 2 (*n* = 35) and >115 for Group 3 (*n* = 34). LDL-c levels were divided based on statistical recommendation to have enough patients in each group. 

Table 1 shows clinical and demographic data according to LDL group. The groups did not differ in clinical or laboratory parameters except for alkaline phosphatase, and platelet, leukocyte and basophil counts, (*p* > 0.05). There were no differences between groups in DM, HTN, IHD, CRP or N-terminal pro b-type natriuretic peptide and other biomarkers.

Table 2 displays comorbidities related to CAD, patients who were admitted with acute myocardial infarction, percutaneous coronary interventions or CABG and the medications used. There were no significant differences between groups, except for recent myocardial infarctions, aortic stenosis due to atherosclerosis and percent of patients on low-dose aspirin therapy.

Figure 1 shows Kaplan–Meier survival rates according to three groups of LDL cholesterol: Group 1 < 80, Group 2, 80–115 and Group 3 > 115 mg/dl. Patients who had LDL ≥ 80 (80–115) and >115 survived longer compared to those with LDL cholesterol ≤ 80 mg/dl (83% vs. 50%) during 18 years of follow-up.

Mortality rates differed among the groups. Patients with LDL < 80 mg/dl had the lowest survival (50%), whereas patients with LDL > 115 mg/dl had 83% survival over 18 years. Patients with LDL > 80 (80–115) and >115 survived longer. However, there was no significant difference between Group 2 and Group 3 patients with LDL cholesterol 80–115 or >115 (*p* = 0.043). We did not find associations between the groups of LDL-c patients and other parameters, including HTN, DM, smoking, creatinine and hemoglobin or other laboratory tests.

We compared the survival rates of patients who received statins (54.7%) and those who did not. The survival rate was lower among patients who received statins (62%) compared to those who were not (78%; *p* = 0.045). 

Figure 2 shows Kaplan–Meier survival rate according to total cholesterol. After adjusting for risk factor-related diseases, HTN, dyslipidemia, smoking, diabetes mellitus, revascularization procedures and NYHA class, patients who had total cholesterol ≤180 had worse prognosis for survival (65%) than those who had total cholesterol >180 mg/dl (84%; *p* = 0.051).

Figure 3 displays Kaplan–Meier survival curve according to triglyceride levels. Higher levels were associated with lower mortality. During the 18-year follow-up, fewer patients with triglyceride level <110 mg/dl survived (48%), in contrast to patients with triglyceride levels >110 mg/dl (83%). We did not find associations between the triglycerides and other parameters, including HTN, DM, smoking, creatinine and hemoglobin or other laboratory tests. This is a new finding which was not reported previously.

No difference in survival was found between patients with HDL > 50 or <50. More males than females had HDL < 50. We did not find associations between the groups of LDL-c patients and other parameters, including HTN, DM, smoking, creatinine and hemoglobin or other laboratory tests.

The same non-significant findings were found regarding the biomarkers oxidized LDL (OX LDL), n-terminal pro brain natriuretic peptide, myeloperoxidase (MPO), heat shock protein (HSP) and CRP.

## 4. Discussion

This study focused on a relatively young (≤51 years old) cohort of patients with HF. The results of this 18-year longitudinal follow-up study indicated that low baseline LDL-c level (regardless of statin use) is a strong prognostic factor of mortality in ischemic HF. This finding supports those of studies that were performed with mostly geriatric patients with HF [9,10,13,17,18,19,23]. This outcome was verified after controlling for established demographic, clinical and laboratory factors and biomarkers of HF survival, including N-terminal pro b-type natriuretic peptide, HSP, MPO, CRP, creatinine, history of diabetes and more, in all three groups of patients. Elevated LDL-c level was found to be positively associated with better long-term survival, while low LDL-c (<80 mg/dl) signified mortality”. Statin therapy for patients with HF who maintained lower LDL-c levels showed a similar trend of increased mortality over the entire study period, as reported previously [5,6,7,8,9,12,13,14,15,23,24,25,26,27,28].

An interesting novel point which was not reported previously is that patients with triglyceride levels <110 mg/dl had lower survival over 18 years of follow-up, in contrast to those with triglyceride levels >110 mg/dl (48% vs. 83%). This may be because HF patients in this group were less balanced and had a stronger catabolic response, which lowers triglyceride levels.

Previous investigations noted this paradox in comparison to ischemic CAD. A study by Horwich et al. [19] reported an inverse relation between total cholesterol and mortality among patients with systolic HF. Many reports on older patients also indicated an association between lower LDL-c levels and poorer survival [28,29].

One explanation for this effect is that in HF, lipids and lipoproteins may have preventive effects by reducing markers of inflammation, such as CRP, cytokines, myeloperoxidase proteinase (MPO), oxidized LDL, interleukin 6 and tumor necrosis factor alpha [18,25,30,31]. Cholesterol can neutralize the liposaccharide and endotoxin components of microbes (such as bacteria that cross the intestine), which are more abundant in advanced HF, and neutralize them by detoxification; thus, reducing inflammation and deactivating these interleukins, cytokines and leukotrienes which are destructive [18,32]. These reports suggested that total cholesterol and LDL-c have a protective role in diverse types of infections. In contrast, HDL levels did not affect survival among HF patients. 

It is important to stress that patients in all three groups were very similar in analyses of main demographics, clinical and laboratory values and biomarkers, except for platelets, which differs from many previous reports [9,26,27,32,33,34]. Body mass index, nutritional status and smoking status of the groups were similar. None of the patients had cardiac cachexia, which could explain low cholesterol levels due to high metabolic rate and low nutritional intake [18].

We believe that the main factor that predicts survival in all kinds of HF is pump failure and not ischemia, which is the end-stage of atherosclerotic CAD. This long-term investigation indicates that low total and LDL-c levels have a predictive effect in decreasing HF. In this relatively young cohort of patients, none had preserved LVEF and we cannot judge the impact of low cholesterol on preserved LVEF. These findings are consistent with recommendations of the European Cardiology Society not to treat HF patients with lipid lowering medications, in contrast to CAD without HF. This should be strongly noted in relation to guidelines that recommend treating patients with low LVEF as an outcome of severe coronary atherosclerosis aggressively with lipid-lowering drugs. Additional research with new lipid lowering drugs, such as PCSK9 and inclisiran is needed to resolve this paradoxical phenomenon. 

Limitations to this study include the relatively small cohort. We did not find significant differences between patients who received statins and those who did not. However, the differences between cholesterol levels were significant, which can be also explained by the small sample size. None of the patients had received new medications such as PCSK9 for CAD, which are different than statins.

## 5. Conclusions

This longitudinal follow-up study showed that among relatively young adults, low LDL-c levels at baseline is a strong predictor of mortality among patients with ischemic HF. This finding is consistent with and confirms previous studies on elderly HF patients. Lipid-lowering therapy is usually not recommended for HF patients with reduced LVEF. Patients with triglyceride levels <110 mg/dl had lower survival during follow-up in comparison to patients with triglyceride level >110 mg/dl. This is a new finding.

## Figures and Tables

**Figure 1 medicina-59-01314-f001:**
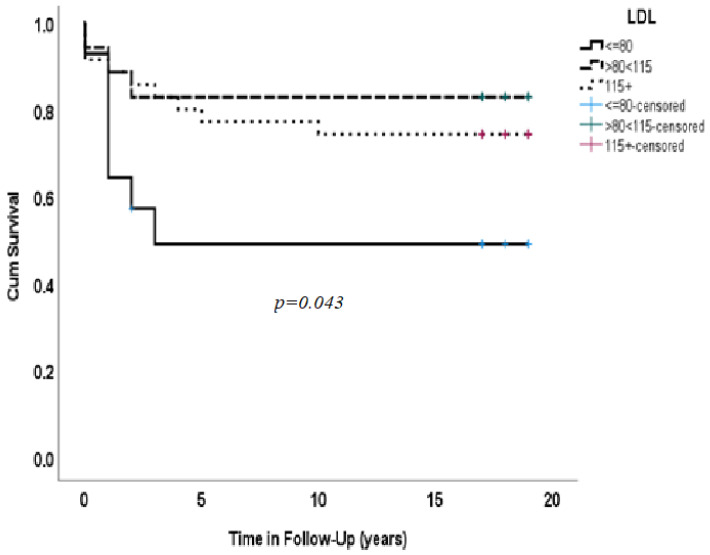
Kaplan–Meier survival rates according to three groups of LDL cholesterol: Group 1 (≤80), Group 2 (80–115) and Group 3 (>115 mg /dl). Patients who had LDL > 80 (80–115) and >115 survived longer (83% vs.) 50% during 18 years of follow-up.

**Figure 2 medicina-59-01314-f002:**
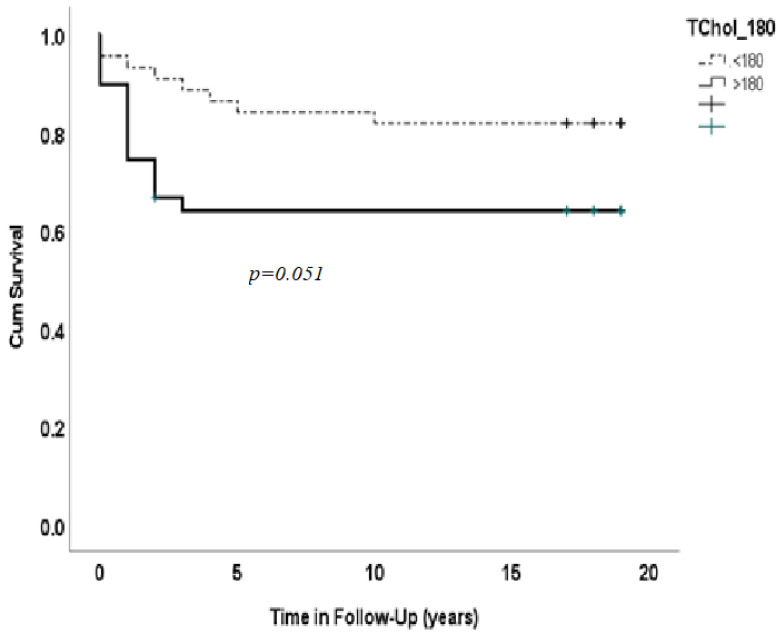
Kaplan–Maier patients survival rate according to total cholesterol ≤180 vs. >180 mg/dl.

**Figure 3 medicina-59-01314-f003:**
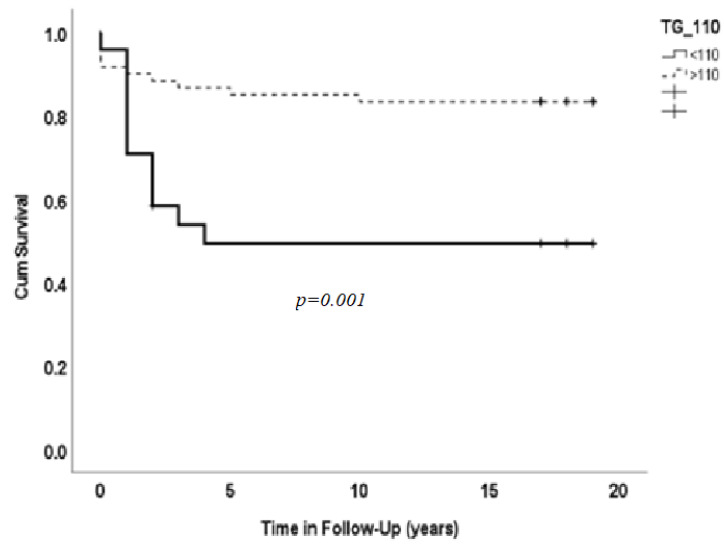
Kaplan–Meier curve of patients’ survival according to triglyceride levels. Higher levels correlated with lower mortality.

**Table 1 medicina-59-01314-t001:** Patient demographics, clinical and laboratory metrics of the LDL groups.

Variable	LDL ≤ 80 *n* = 15		>80 LDL < 115 *n* = 35		LDL > 115 *n* = 34		*p*-Value *
	Mean	SD	Mean	SD	Mean	SD	
Age in years	48.7	2.3	48.5	3.4	49.0	3.0	0.783
Myeloperoxidase	417.3	935.7	208.2	200.6	179.0	166.2	0.186
Hb, g%	12.4	1.2	13.3	1.4	13.3	1.5	0.098
Glucose, mg/dl	135.1	61.1	131.2	57.4	142.4	79.5	0.803
BUN, mg/dl	35.3	21.5	32.7	26.4	32.7	18.1	0.920
Sodium, mg/dl	139.3	4.3	139.6	3.7	138.4	4.5	0.500
Potassium, mg/dl	4.7	0.6	4.5	0.5	4.8	0.7	0.131
Chloride, mg/dl	100.4	4.8	100.6	4.5	101.1	4.8	0.857
Creatinine mg/dl	1.7	1.1	1.5	0.8	1.9	1.3	0.312
Creatinine clearance	70.5	44.3	80.2	29.1	70.0	26.3	0.360
Total cholesterol, mg/dl	155.4	30.2	173.1	28.8	218.6	34.1	0.000 ** 3 ≠ 1, 2
Calcium, mg/dl	9.3	0.7	9.4	0.6	9.5	0.7	0.527
Uric acid, mg/dl	7.5	2.4	7.4	2.7	8.1	2.4	0.530
Phosphor, mg/dl	3.9	0.5	5.3	7.0	4.0	0.7	0.449
LDH, mg/dl	329.9	104.1	325.0	95.2	326.3	90.1	0.987
CPK, mg/dl	59.6	38.0	70.7	40.9	64.3	49.0	0.696
GOT	23.5	9.4	21.6	9.8	21.8	6.0	0.749
Alkaline phosphatase, mg/dl	87.9	71.9	53.7	20.9	62.9	34.9	0.027 **
Total protein, g/l	72.1	4.0	66.6	16.4	63.7	19.0	0.256
Albumin	41.5	2.5	39.4	9.9	36.8	12.8	0.314
GPT	21.5	14.2	21.2	13.9	25.6	13.8	0.400
Bilirubin, mg/dl	1.0	0.9	0.7	0.4	0.7	0.3	0.062
WBC * 1000	7.7	2.8	6.8	1.5	8.4	2.5	0.017 ** 2 ≠ 3
Polymorphonuclear cells %	61.8	19.2	62.0	7.4	66.0	8.9	0.298
Eosinophiles%	3.5	1.5	3.1	2.0	2.5	1.7	0.217
Basophiles %	1.1	0.9	0.6	0.4	0.5	0.3	0.010 ** 1 ≠ 2, 3
Lymphocytes%	24.6	7.3	28.3	12.2	24.8	10.2	0.408
Monocytes%	6.7	1.6	7.7	4.2	8.0	11.2	0.902
Platelets * 1000	182.1	24.9	248.5	64.0	235.4	81.5	0.033 ** 1 ≠ 2

* One-Way ANOVA or Kruskal–Wallis, as appropriate. ** Differences by Bonferroni.

**Table 2 medicina-59-01314-t002:** Patient comorbidities according to LDL group.

Comorbidities	LDL ≤ 80 (*n* = 15)	>80 LDL < 115 (*n* = 35)	LDL > 115 (*n* = 34)	*p*-Value
	*n* (%)	*n* (%)	*n* (%)	
Hypertension	7 (46.7)	21 (60)	18 (52.9)	0.660
Diabetes mellitus	7 (46.7)	12 (34.3)	11 (31.4)	0.579
IHD/MI Smoking	15 (100) 6(40.0)	34 (97.1) 13(37.1)	28 (28.8) 13(38.2)	0.019 0.36
Aortic stenosis	1 (6.7)	1 (2.9)	9 (25.7)	0.013
Chronic atrial fibrillation (CAF)	2 (13.3)	3 (8.6)	5 (14.3)	0.743
TIA/CVA	2 (13.3)	1 (2.9)	2 (5.7)	0.353
PTCA/CABG	6 (40)	16 (45.7)	12 (34.3)	0.621
Medications				
Coumadin	4 (26.7)	4 (11.4)	10 (28.6)	0.182
Aspirin	7 (46.7)	27 (77.1)	17 (48.6)	0.026
Statins	6 (40.0)	21 (60.0)	19 (54.3)	0.429
ACE inhibitor	5 (33.3)	20 (57.1)	15 (42.9)	0.245
ARB	3 (20.0)	10 (28.6)	6 (17.1)	0.503
Plavix	0	2 (5.7)	1 (2.9)	0.581
Nitrates	3 (20.0)	9 (25.7)	6 (17.1)	0.675
Ca-block	4 (26.7)	3 (8.6)	5 (14.3)	0.242
Beta-blockers	9 (60.0)	26 74.3()	19 (54.3)	0.210
Insulin	5 (6.7)	4 (11.4)	1 (2.9)	0.375
Oral hypoglycemics	4 (26.7)	7 (20.0)	6 (17.1)	0.743
Alpha-blockers	2 (13.3)	5 (14.3)	2 (5.7)	0.472
Bezafibrate	1 (6.7)	7 (20.0)	3 (8.6)	0.264
Antiarrhythmics	4 (26.7)	4 (11.4)	5 (14.3)	0.381
Digoxin	4 (26.7)	6 (17.1)	11 (31.4)	0.376
Spironolactone	8 (53.3)	19 (54.3)	16 (45.7)	0.752
Diuretics	11 (73.3)	26 (74.3)	26 (74.3)	0.997

All 15 patients who presented with with MI had LDL-c < 80 mg/dl. There were no correlations between LDL-c, total cholesterol and chronic atrial fibrillation (CAF), CVA/TIA, CABG and medications, including statins.

## Data Availability

The study data may be obtained upon request to the corresponding author.

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
