# Peer review of "Low Cholesterol Levels in Younger Heart Failure Patients May Predict Unfavorable Outcomes"

_medicina, 2023, doi:10.3390/medicina59071314_

Round 1
Reviewer 1 Report
The present manuscript focuses on a very important, frequent and everyday health problem concerning low cholesterol levels in younger heart failure patients and prognosis on survival. This longitudinal cohort study has shown a negative association between LDL-c levels and mortality among those patients with heart failure, similar findings with previous studies in older patient groups.
However, if researchers might relatively increase the sample size in order to better and more accurately compare statin effects and other parameters in each group and add some other newer pro-inflammatory and preclinical atherosclerotic biomarkers as albuminuria, VEGF-A, TGF-β1, FGF-21 and -23, MMP-2 and -9, TIMP-1 and -2, isoprostane-8 and -15, ox-LDL, IL-6, carotid IMT(intima-media thickness) in order to better describe this very important clinical model.
I personnally would like to see a revised form of this manuscript on this direction in order to provide to the literature a state of the art.
Reviewer 2 Report
In this study authors included outpatient HF patients (<51 years of age) between Jan 1998 to July 2001. The mean follow-up was 11.3 years. included patients with LVEF <40%. They compared outcomes between three LDL groups: <80, 80-115 and >115.
Major revisions needed:
Can authors provide details about patient's smoking status in table 2? Current, past or never smoker?
In group with patient with LDL <80 ng/dl:There were more patients with ischemic heart disease and diabetes. Only 46% took aspirin, 40% took statin and very few were on GDMT. Can authors analysis to adjust for comorbidities and GDMT use and then evaluate if LDL-C <80 is independently associated with mortality?
Why was cox regression only used for Total cholesterol to adjust for comorbidities and not with LDL-C, Triglycerides?
Do authors have information regarding CKD stage of the patients? If yes, that should also be included in the adjustment model?
In table 2, in beta blocker row under LDL <80 column it should be 60% instead of 6%. and some percentage are out of parenthesis like spironolactone. please correct. Please double check all values throughout the manuscript for accuracy.
In discussion " Elevated LDL-c level was found as the most significant independent determinate of better long-term survival, while low LDL-c (<80 mg/dl).... were significant predictor of mortality". I don't think this statement is accurate. The authors did not show adjusted analysis results for LDL-C in results section.
Minor comments:
In Results author stated: "The mean follow-up was 11.3 years (range 38-51 years)." I doubt that number in parenthesis is standard deviation. what does range 38-51 means. I would suggest authors to write standard deviation in parenthesis.
What does CAF stand for in the table? Please expand all the abbreviations at first use. Also expand abbreviations used in table.
Page 4, line number 135: spelling of KM curve is mis-spelled. It should be "Kaplan-Meier". Similarly spelling needs to be corrected in figure 1, 2 and 3. Needs to be rectified throughout the manuscript.
Authors commented in Introduction section that "several other studies showed paradoxical results with decreased total and LDL-c (especially through statin treatment),indicating the possibility that they may be harmful for patients with HF." Can they please provide citation for this?
This sentence in discussion needs citation: "It supports the results of previous studies that were performed on mostly geriatric patients with HF."
There are grammatical errors and spelling mistake throughout the manuscript. Recommend a thorough read to rectify those.
Author Response
In this study authors included outpatient HF patients (<51 years of age) between Jan 1998 to July 2001. The mean follow-up was 11.3 years. included patients with LVEF <40%. They compared outcomes between three LDL groups: <80, 80-115 and >115.
Major revisions needed:
Can authors provide details about patient's smoking status in table 2? Current, past or never smoker?
9.3% of the patients were smokers (active or past) row-105
In group with patient with LDL <80 ng/dl: There were more patients with ischemic heart disease and diabetes. Only 46% took aspirin, 40% took statin and very few were on GDMT. Can authors analysis to adjust for comorbidities and GDMT use and then evaluate if LDL-C <80 is independently associated with mortality?
In previous studies we showed that low LDL-c was associated with mortality (adjusted to other comorbidities and medicines). This is a retrospective study which started more 25 years ago (1998). There were no established GDMT nor by statins treatment guidelines especially for HF patients.
Why was cox regression only used for Total cholesterol to adjust for comorbidities and not with LDL-C, Triglycerides?
Review of the data with our senior statistician showed that Kaplan Meier survival analysis better fits our results. Therefore, a figure depicting Kaplan Meyer survival was substituted for figure 2.
Do authors have information regarding CKD stage of the patients? If yes, that should also be included in the adjustment model?
Yes, it was taken and adjusted. We did not find associations between the group of LDL-c patients and other parameters, including HTN, DM, smoking, creatinine and hemoglobin or other laboratory tests. Row-168
In table 2, in beta blocker row under LDL <80 column it should be 60% instead of 6%. and some percentage are out of parenthesis like spironolactone. please correct. Please double check all values throughout the manuscript for accuracy.
We corrected the table 2.
In discussion " Elevated LDL-c level was found to be positively associate with better long-term survival, while low LDL-c (<80 mg/dl).... were significant predictor of mortality". I don't think this statement is accurate. The authors did not show adjusted analysis results for LDL-C in results section.
This statement was corrected 181
Minor comments:
In Results author stated: "The mean follow-up was 11.3 years (range 38-51 years)." I doubt that number in parenthesis is standard deviation. what does range 38-51 means. I would suggest authors to write standard deviation in parenthesis.
Follow up was 38-51 years with a median of 11.3 years. This was corrected in the text.
What does CAF stand for in the table? Please expand all the abbreviations at first use. Also expand abbreviations used in table.
CAF- is chronic atrial fibrillation . We spelled it in the table
Page 4, line number 135: spelling of KM curve is mis-spelled. It should be "Kaplan-Meier". Similarly spelling needs to be corrected in figure 1, 2 and 3. Needs to be rectified throughout the manuscript.
We spelled KM as Kaplan – Meyer and corrected them in the figures
Authors commented in Introduction section that "several other studies showed paradoxical results with decreased total and LDL-c (especially through statin treatment),indicating the possibility that they may be harmful for patients with HF." Can they please provide citation for this?
The citations were provided[5–9,11–15,17-22]
This sentence in discussion needs citation: "It supports the results of previous studies that were performed on mostly geriatric patients with HF."
We added citation[9,10,13,17-19,23
Comments on the Quality of English Language
There are grammatical errors and spelling mistake throughout the manuscript. Recommend a thorough read to rectify those.
To eliminate spelling and grammar errors as well as reducing repetition rate to a minimum, the manuscript was reviewed and edited by Mrs. Faye Schreiber, MA, who is qualified in medical copyediting.
please see the attachment

Round 2
Reviewer 2 Report
Authors have improved manuscript after incorporating the changes.
Major:
Only one thing I am still not sure why the authors did not adjust for risk factor-related diseases: HTN, dyslipidemia, smoking, diabetes mellitus, revascularization procedures and NYHA class for LDL-c and TG as they did for Total cholesterol in KM curve. Thanks for changing the wording in discussion.
Minor: still need to expand on lot of abbreviations used in tables.
Author Response
Respond to Reviewer
Authors have improved manuscript after incorporating the changes.
Only one thing I am still not sure why the authors did not adjust for risk factor-related diseases: HTN, dyslipidemia, smoking, diabetes mellitus, revascularization procedures and NYHA class for LDL-c and TG as they did for Total cholesterol in KM curve. Thanks for changing the wording in discussion.
We adjusted risk factors for Total cholesterol, LDL-cholesterol and triglycerides however didn‘t mentioned them in the text in previous version.
Our senior statistician made recalculations and the were no differences in survival after adjustment pages 9 and 10 (now they written in red)
Minor: still need to expand on lot of abbreviations used in tables.
We expand the abbreviations in tables
Thank you very much